Determining relevant traits for selecting landrace accessions of Phaseolus lunatus L. for insect resistance

Ruiz-Santiago Roberto Rafael 1
Ballina-Gómez Horacio Salómon horacio.bg@conkal.tecnm.mx 1
Ruiz-Sánchez Esau 1
Martínez-Castillo Jaime 2
Garruña-Hernández René 3
Andueza-Noh Rubén Humberto 3
1 Division de Estudios de Posgrado e Investigacion, Tecnologico Nacional de México/Campus Conkal , Conkal , Yucatan , Mexico
2 Centro de Investigacion Cientifica de Yucatan , Merida , Yucatan , Mexico
3 Division de Estudios de Posgrado e Investigacion, Conacyt-Tecnológico Nacional de México/Campus Conkal , Conkal , Yucatan , Mexico
Kalaji Hazem
Electronic publication date: 2021 Sep 16
Publication date: 2021
Volume: 9
Electronic Location ID: e12088
Received 2021 Feb 25; Accepted 2021 Aug 8
Copyright: ©2021 Ruiz-Santiago et al.
Copyright year: 2021
Copyright holder: Ruiz-Santiago et al.
License: This is an open access article distributed under the terms of the Creative Commons Attribution License, which permits unrestricted use, distribution, reproduction and adaptation in any medium and for any purpose provided that it is properly attributed. For attribution, the original author(s), title, publication source (PeerJ) and either DOI or URL of the article must be cited.
License URL: https://creativecommons.org/licenses/by/4.0/

Keywords: Lima bean, Plant defense, Leaf damage, Seed yield, Plant physiology

Funding: CONACYT 845968 YUC-2018-03-01-119959 Tecnológico Nacional de México (TECNM) project 8983.20-P Selección de variedades locales con características morfológicas y de defensa en contra de insectos herbívoros en el cultivo de Phaseolus lunatus L This work was supported by CONACYT with the doctoral scholarship to Roberto Rafael Ruiz-Santiago (No. 845968), CONACYT project (no. YUC-2018-03-01-119959) and Tecnológico Nacional de México (TECNM) project (no. 8983.20-P): “Selección de variedades locales con características morfológicas y de defensa en contra de insectos herbívoros en el cultivo de Phaseolus lunatus L.”. The funders had no role in study design, data collection and analysis, decision to publish, or preparation of the manuscript.

==============================
Plant-insect interactions are a determining factor for sustainable crop production. Although plants can resist or tolerate herbivorous insects to varying degrees, even with the use of pesticides, insects can reduce plant net productivity by as much as 20%, so sustainable strategies for pest control with less dependence on chemicals are needed. Selecting plants with optimal resistance and photosynthetic traits can help minimize damage and maintain productivity. Here, 27 landrace accessions of lima beans, Phaseolus lunatus L., from the Yucatan Peninsula were evaluated in the field for morphological resistance traits, photosynthetic characteristics, insect damage and seed yield. Variation was found in physical leaf traits (number, area, and dry mass of leaves; trichome density, specific leaf thickness and hardness) and in physiological traits (photosynthetic rate, stomatal conductance, intercellular carbon, water-use efficiency, and transpiration). Five accessions (JMC1325, JMC1288, JMC1339, JMC1208 and JMC1264) had the lowest index for cumulative damage with the highest seed yield, although RDA analysis uncovered two accessions (JMC1339, JMC1288) with strong positive association of seed yield and the cumulative damage index with leaf production, specific leaf area (SLA) and total leaf area. Leaf traits, including SLA and total leaf area are important drivers for optimizing seed yield. This study identified 12 important morphological and physiological leaf traits for selecting landrace accessions of P. lunatus for high yields (regardless of damage level) to achieve sustainable, environmentally safe crop production.

Introduction

In recent years, global crop productivity has been improved through the artificial selection of traits that increase yield (Lynch, 2007), but this approach has resulted in low levels of diversity, lack of expression of defense genes, and production systems that depend on high inputs of pesticides (Panda & Khush, 1995; Lynch, 2007; Chen et al., 2016). One of the main constraints on crops is insect damage to leaves; up to 20% of the net productivity in important crops can be lost despite increased pesticide use (Oerke & Dehne, 2004; Agrawal, 2011). Stout (2013) proposed an effective sustainable alternative: selecting plants with resistance and tolerance traits that reduce the impact of insect damage. The efficacy of such resistance traits in protecting a crop against herbivorous insects depends on studies that clarify their influence, their impact on herbivorous insects and their expression under different environmental conditions (Strauss & Agrawal, 1999; Stinchcombe, 2002). Some accessions with larger leaves are likely to have greater vigor and tolerance to damage by herbivorous insects (Ssekandi et al., 2016). Likewise, the thickness and hardness of leaves also have an important impact on resistance. In Fabaceae species, trichomes can also contribute to resistance against defoliating insects (Oghiakhe, Jackai & Makanjuola, 1992; Veeranna & Hussain, 1997) by hindering their movement on the plants (Tian et al., 2012; Figueiredo et al., 2013). In contrast, some plants with thinner, more fragile leaves are less preferred by insects, as in the case of Vigna radiata (L.) (Lakshminarayan, Singh & Mishra, 2008), V. mungo (Taggar & Gill, 2012), Gossypium hirsutum (Butter & Vir, 1989) and Cucumis sativus (Shibuya et al., 2009).

Legumes are the second most important group in past and current agricultural systems and for human nutrition (Blair et al., 2016). According to the FAO in 2018, beans were third in importance by planted area in Mexico, with 7.9% of the total. The Yucatan Peninsula has the greatest richness of cultivated domesticated beans in all of Mexico, and varieties have high levels of genetic diversity including those of P. lunatus (Martínez-Castillo et al., 2012), which are excellent germplasm sources for improving cultivated beans. We hypothesized that landrace accessions of broad beans (Phaseolus lunatus) have physical and physiological characteristics that confer defensive traits that may reduce the activity of herbivorous insects and optimize yields. In this context, the objective of this study was to identify accessions that were most resistant to damage caused by defoliating insects in 27 landrace accessions of lima beans (P. lunatus) by determining morphological, physiological and yield traits known to contribute to insect resistance such as dry mass of leaves, leaf area, specific leaf area (SLA), leaf thickness and hardness, and trichome density, improvement in photosynthetic capacity, and its impact on yield (Gong & Zhang, 2014).

Materials and Methods

Seed sources

Seeds of 27 landraces accessions of lima beans (P. lunatus) were collected in the states of Quintana Roo, Campeche and Yucatán in the Yucatan peninsula from home gardens and rural markets. Information on the origin and genetic characterization of landraces is available in previous studies carried out by the Centro de Investigacion Cientifica de Yucatan (CICY), in Martínez-Castillo, Colunga-García & Zizumbo-Villarreal (2008) and Camacho-Pérez, Martínez-Castillo & Mijangos-Cortés (2008) (Table 1). Seeds were tested for germination, and only those lots with germination above 85% were sown.

Table 1 Origin of landrace accessions of lima bean (P. lunatus) evaluated in this study.

Accesion Code	Accession	Species	Collector	State	Municipality	Coordinate	Local name	
1	RRS0001	Phaseolus lunatus	Roberto Ruiz	Yucatán	Izamal	–	–	
2	JMC1271	Phaseolus lunatus	Jaime Martínez	Quintana Roo	Tulum	87°46′16.55″	Putsicasutsuy	
3	JMC1280	Phaseolus lunatus	Jaime Martínez	Quintana Roo	Felipe C. Puerto	88°02′43″	Mulición	
4	RRS0002	Phaseolus lunatus	Roberto Ruiz	Yucatán	Izamal	–	–	
5	JMC1255	Phaseolus lunatus	Jaime Martínez	Campeche	Calkiní	90°03′03″	Mulición	
6	JMC1304	Phaseolus lunatus	Felix Dzul Tejero	Campeche	Calkiní	90°03′03″	Mulición	
7	JMC1240	Phaseolus lunatus	Jaime Martínez	Yucatán	Peto	89°24′00″	Putsicasutsuy	
8	JMC1350	Phaseolus lunatus	Jaime Martínez	Campeche	Hopelchén	89°44′51.98″	X-Nuk ib	
9	JMC1254	Phaseolus lunatus	Jaime Martínez	Yucatán	Tahdziú	89°30′	Putsicasutsuy	
10	JMC1327	Phaseolus lunatus	Jaime Martínez	Campeche	Hecelchakán	89°58′14.48″	Mulición	
11	JMC1273	Phaseolus lunatus	Jaime Martínez	Campeche	Hopelchén	89°35.57′	Putsicasutsuy	
12	JMC1357	Phaseolus lunatus	Jaime Martínez	Campeche	Calkiní	89°53′90″	Mejen ib	
13	JMC1345	Phaseolus lunatus	Jaime Martínez	Yucatán	Tixmehuac	89°06′31.43″	Mulición	
14	JMC1270	Phaseolus lunatus	Jaime Martínez	Yucatán	Tixmehuac	89°6′56.16″	Chak ib	
15	JMC1337	Phaseolus lunatus	Jaime Martínez	Campeche	Hopelchén	89°11′30″	Mulición	
16	JMC1245	Phaseolus lunatus	Felix Dzul Tejero	Campeche	Calkiní	89°53′90″	Putsicasutsuy	
17	JMC1208	Phaseolus lunatus	Jaime Martínez	Yucatán	Chankom	88°30′48.00″	Sac ib	
18	JMC1348	Phaseolus lunatus	Jaime Martínez	Yucatán	Peto	89°24′00″	Sak X-nuk ib	
19	JMC1339	Phaseolus lunatus	Jaime Martínez	Yucatán	Yaxcabá	88°49′39.69″	Sac ib	
20	JMC1288	Phaseolus lunatus	Felix Dzul Tejero	Yucatán	Tekax	89°29′18″	Box ib	
21	JMC1306	Phaseolus lunatus	Jaime Martínez	Yucatán	Tzucacab	89°57′35″	Mulición	
22	JMC1264	Phaseolus lunatus	Jaime Martínez	Yucatán	Yaxcabá	88°49′39.69″	Chak ib	
23	JMC1325	Phaseolus lunatus	Jaime Martínez	Yucatán	Tekax	89°17′16″	Mejen ib	
24	JMC1297	Phaseolus lunatus	Felix Dzul Tejero	Yucatán	Tixmehuac	89°6′56.16″	Sac ib	
25	JMC1313	Phaseolus lunatus	Jaime Martínez	Yucatán	Tekax	89°12′00″	Sacmejen	
26	JMC1336	Phaseolus lunatus	Jaime Martínez	Yucatán	Tekax	89°17′16″	Sacmejen	
27	JMC1364	Phaseolus lunatus	Jaime Martínez	Yucatán	Tekom	88°27′17″	Sac ib	

Site conditions and crop establishment

The field experiment was carried out in September, October, November and December of 2019 in the horticultural production area of the National Technological Institute of Mexico, Campus Conkal when the monthly mean temperature was 26.4 °C, the maximum was 34.7 °C, and minimum was 17.3 °C, and the monthly mean precipitation was 100.66 mm. The soil is a Leptosol, with 0.93% N, and the total contents of P, K, Ca and Mg is 2.45, 3.5, 49.38 and 2.63 g kg−1, respectively. Seeds were sown directly every 60 cm in a 50-m row with a distance of 120 cm between each row. Approximately 100 plants per accession were obtained. The field was irrigated each day (7:00 to 9:00) with a drip irrigation strip system, and traditional agronomic management was applied for weed control; no pesticides or chemical fertilizers were applied. Plots were established using a split-plot experimental design with a completely random arrangement of three subplots.

Data collection

At 60 days after plant emergence (DAE), morphological and physiological leaf traits of each landrace accession were evaluated. We randomly selected five plants for each of three experimental subplots for each accession for a total of 15 plants. All leaves on each plant were counted, and area of each leaf was measured with an area meter (LI-3000C portable meter, LI-COR, Lincoln NE, USA). The dry mass of leaves was obtained by placing fresh leaves in a drying oven at 60 °C until mass was constant. The SLA (cm2 g−1) was calculated by dividing the leaf area (cm2) by the dry mass (g). Leaf thickness was measured with a digital micrometer (Mitutoyo model H-2780 JPN). Blade hardness (g cm−2) was measured with a portable penetrometer (AMS 59032 OSHA, USA). Trichomes were counted on several parts of the adaxial surface of the fully expanded youngest leaf using a stereoscope (OPTIKA ST-30FX IT) at 40 × to calculate density (no. cm−2) (Widstrom, McMillian & Wiseman, 1979).

An infrared gas analyzer (LI-6400 IRGA; LI-COR, Lincoln NE, USA) was used to measure gas exchange in three fully extended young leaves for each of 15 new randomly selected plants of each accession. Each leaf was measured five times for photosynthetic assimilation rate (PN), stomatal conductance (gs), intercellular carbon (Ci), and transpiration (E). Water-use efficiency (WUE) was then calculated as photosynthetic assimilation rate (PN) divided by transpiration rate (E). Measurements were done between 7:00 and 10:00 when flowering had started (between 45 and 60 DAE).

Leaf damage and yield

To evaluate foliar damage by chewing insects on 15 plants randomly selected per plot at 30, 45 and 60 DAE, we used the percentage damage scale of Dirzo & Domínguez (1995): (0) leaves without herbivory, (1) 1 to 5% damage, (2) 6 to 12% damage, (3) 13 to 25% damage; (4) 26 to 50% damage and (5) above 50% damage). At each sampling date, we also calculated a cumulative damage index for each accession by dividing the lowest damage value by the highest value for the accession (Sohrabi, Nooryazdan & Gharati, 2017).

For yield determinations, 15 plants were selected per accession, pods were harvested and beans removed and oven-dried at 60 °C until mass was constant. Beans were then weighted to obtain seed yield per plant (g plant−1) for each accession.

Statistical analyses

We ran a one-way analysis of variance (ANOVA) to compare morphological and physiological traits, damage and seed yield among the accessions; thus, we used accession as the only factor (independent variable). When the data did not meet normality assumptions, data were transformed as follows: continuous data with the natural logarithm, discrete numbers with square root, and proportions with the arcsine of the square root. Hierarchical grouping of means tests of the Scott-Knott statistic were then applied.

These analyses were performed with the InfoStat software (Di Rienzo et al., 2016). The most important resistance and physiological tolerance traits for the accessions were identified using a principal component analysis (PCA) and the arithmetic means for each accession for a variable. The PCA was performed using correlation matrices and normalization of the model by the varimax rotation method (Dien, Beal & Berg, 2005) using SPSS version 25 (IBM, Armonk, NY, USA).

To determine possible associations between variables, we subjected all data for physical and physiological traits, seed yield and cumulative damage index for each of the 27 accessions using a redundancy analysis (RDA) (Legendre & Legendre, 1998). The RDA was chosen over a canonical correspondence analysis due to the length of the gradient for the variables (Ter Braak & Smilauer, 2002). Gradient length was calculated using a detrended correspondence analysis (DCA) (Hill, 1979). The significance of the damage and yield index on the ordering of morphological and physiological variables, was analyzed using a Monte Carlo random permutation test (499 permutations, p < 0.05) using Canoco 4.5 (Ter Braak & Smilauer, 2002).

Results

Physical traits of resistance

Significant differences in the physical traits were found among the accessions (Scott-Knott p < 0.005). The accessions with the most leaves were JMC1339, JMC1306, JMC1364, JMC1208, JMC1264, JMC1313, JMC1336, JMC1288 and JMC1348 had the fewest (Fig. 1A). JMC1339 had the largest leaf area, JMC1255 and JMC1348 the smallest (Fig. 1B). JMC1255, RRS0002, JMC1280 JMC1348 and JMC1304 had the highest leaf dry mass; JMC1339 had the lowest (Fig. 1C). JMC1255 had the thickest leaves, and JMC1325 accession had the thinnest (Fig. 1D). JMC1339 had the highest SLA, and JMC1255, JMC1348, RRS002, JMC1280, JMC1304 and JMC1357 had the lowest (Fig. 1E). The highest trichome densities were found on JMC1306 and JMC1280, the lowest on JMC1208 (Fig. 1F). JMC1264, JMC1325 and JMC1313 had the hardest leaves, and the rest had softer leaves (Fig. 1G).

Figure 1 Means (±SD) at 60 days after emergence for leaf resistance traits of 27 Phaseolus lunatus landrace accessions from southeastern Mexico.

Different letters above histobars denote a significant difference among accessions (Scott-Knott, p < 0.05). (A) Number of leaves, (B) leaf area, (C) dry mass of leaves, (D) thickness, (E) specific leaf area, (F) number of trichomes, (G) hardness. Accession codes on x-axis: 1 = RRS0001, 2 = JMC1271, 3 = JMC1280, 4 = RRS0002, 5 = JMC1255, 6 = JMC1304, 7 = JMC1240, 8 = JMC1350, 9 = JMC1254, 10 = JMC1327, 11 = JMC1273, 12 = JMC1357, 13 = JMC1345, 14 = JMC1270, 15 = JMC1337, 16 = JMC1245, 17 = JMC1208, 18 = JMC1348, 19 = JMC1339, 20 = JMC1288, 21 = JMC1306, 22 = JMC1264, 23 = JMC1325, 24 = JMC1297, 25 = JMC1313, 26 = JMC1336, 27 = JMC1364. Days after emergence (DAE), specific leaf area (SLA).

Physiological traits

The physiological variables also differed significantly among the accessions (Scott-Knott p < 0.005). Accessions JMC1273, JMC1325, JMC1264, RRS0002, JMC1245, JMC1288, JMC1313, JMC1304, JMC1280, JMC1357, JMC1270 and JMC1350 had the highest photosynthetic rate (PN), JMC1336 and JMC1339 the lowest (Fig. 2A). WUE was highest in JMC1337, JMC1245, and JMC1270, and lowest in JMC1336 (Fig. 2B). JMC1364 had the highest gs, for overall (Fig. 2C). Transpiration was highest in JMC1273 and JMC1364 (Fig. 2D). JMC1336 and JMC1364 had the highest Ci for overall (Fig. 2E).

Figure 2 Means (±SD) at 60 days after emergence for physiological tolerance traits of 27 Phaseolus lunatus landraces accessions from southeastern Mexico.

Different letters above histobars denote a significant difference among accessions at 60 DAE (Scott-Knott, p < 0.05). (A) Photosynthesis assimilation rate (PN), (B) water-use efficiency (WUE), (C) stomatal conductance (gs), (D) transpiration (E), (E) intercellular carbon (Ci). Accession codes on x-axis: 1 = RRS0001, 2 = JMC1271, 3 = JMC1280,4 = RRS0002, 5 = JMC1255, 6 = JMC1304, 7 = JMC1240, 8 = JMC1350, 9 = JMC1254, 10 = JMC1327, 11 = JMC1273, 12 = JMC1357, 13 = JMC1345, 14 = JMC1270, 15 = JMC1337, 16 = JMC1245, 17 = JMC1208, 18 = JMC1348, 19 = JMC1339, 20 = JMC1288, 21 = JMC1306, 22 = JMC1264, 23 = JMC1325, 24 = JMC1297, 25 = JMC1313, 26 = JMC1336, 27 = JMC1364.

Leaf damage and yield

The percentage of leaf damage differed significantly among all accessions (Scott-Knott p < 0.005) and at DAE (Scott-Knott p < 0.005). The highest percentages of damage were found at 45 DAE, followed by 60 and 30 DAE. At 30 DAE, accessions JMC1271, JMC1255 and RRS0001 had the highest percentages of damage, and JMC1325, JMC1336, JMC1297, JMC1313, JMC1288, JMC1270, JMC1364, JMC1348, JMC1337, JMC1245, JMC1339, JMC1306 and JMC1264 had the lowest (Fig. 3A). For the damage index, RRS0001, JMC1271 and JMC1255 had the highest values; JMC1297, JMC1325, JMC1306, JMC1273 and JMC1264 had the lowest (Table 2). At 45 DAE, JMC1273 had the highest percentage of foliar damage; JMC1306, JMC1264 and JMC1357 had the lowest (Fig. 3A). The damage index at 45 DAE showed that JMC1255, JMC1273, JMC1280 and JMC1339 had the greatest damage; JMC1306, JMC1264, JMC1325 and RRS0002 had the lowest (Table 2). At day 60 DAE, the highest percentages of leaf damage were on JMC1348, JMC1336, JMC1339, JMC1288, JMC1327, the lowest were on JMC1297, JMC1357, JMC1306, JMC1304, JMC1255 and JMC1325 (Fig. 3C). In addition, the damage index was notably higher for JMC1336 than for JMC1348, JMC1339, JMC1288, JMC1306, JMC1264, JMC1325, JMC1297 and JMC1313, which had the lowest values (Table 2). Seed yields also differed significantly among accessions (Scott-Knott p < 0.005), the accessions JMC1325 and JMC1348 had the highest yield (Table 2).

Figure 3 Mean (±SD) leaf damage caused by herbivorous insects at three ages of 27 Phaseolus lunatus landrace accessions from southeastern Mexico.

Different letters above histobars denote a significant difference among accessions at (A) 30, (B) 45 and (C) 60 days after emergence (DAE) (Scott-Knott, p < 0.05).

Table 2 Estimates of cumulative damage index at three growth ages (30, 45 and 60 days after emergence; DAE) and of seed yield at 60 DAE for lima bean (P. lunatus).

The index was calculated by dividing the lowest recorded number by the highest recorded number for each accession, at 30 45 y 60 DAE (days after emergence). For seed yield, different letters within a column denote a significant difference among accessions within the each DAE (Scott-Knott, p < 0.05). Means (±SD).

Accession code	Accessions	Damage index
30 DAE	Damage index
45 DAE	Damage index
60 DAE	Seed yield (g plant −1 )		
1	RRS0001	0.14	0.24	0.14	53.6 ± 0.62	c	
2	JMC1271	0.11	0.03	0.11	49.7 ± 0.93	c	
3	JMC1280	0.03	0.38	0.09	53.6 ± 0.62	c	
4	RRS0002	0.02	0.02	0.21	49.4 ± 0.74	d	
5	JMC1255	0.14	0.42	0.04	38.5 ± 0.61	d	
6	JMC1304	0.07	0.07	0.18	33.0 ± 0.75	e	
7	JMC1240	0.06	0.11	0.23	28.1 ± 0.92	e	
8	JMC1350	0.06	0.04	0.24	34.5 ± 0.77	e	
9	JMC1254	0.03	0.09	0.11	31.3 ± 0.92	e	
10	JMC1327	0.04	0.03	0.21	34.9 ± 0.74	e	
11	JMC1273	0.01	0.42	0.21	58.8 ± 1.69	b	
12	JMC1357	0.02	0.11	0.03	35.0 ± 0.74	e	
13	JMC1345	0.03	0.14	0.03	32.8 ± 0.61	e	
14	JMC1270	0.03	0.22	0.23	29.0 ± 0.92	e	
15	JMC1337	0.03	0.08	0.25	38.4 ± 0.92	d	
16	JMC1245	0.03	0.08	0.07	30.8 ± 0.84	e	
17	JMC1208	0.03	0.16	0.03	52.4 ± 0.01	b	
18	JMC1348	0.05	0.05	0.18	66.6 ± 0.92	a	
19	JMC1339	0.02	0.33	0.02	54.9 ± 0.92	c	
20	JMC1288	0.09	0.34	0.02	55.9 ± 2.24	c	
21	JMC1306	0.01	0.00	0.03	42.4 ± 0.43	d	
22	JMC1264	0.01	0.00	0.02	51.9 ± 0.92	c	
23	JMC1325	0.00	0.02	0.02	67.0 ± 0.92	a	
24	JMC1297	0.00	0.02	0.06	30.8 ± 0.92	e	
25	JMC1313	0.02	0.20	0.03	39.5 ± 0.92	d	
26	JMC1336	0.00	0.04	0.59	42.1 ± 0.74	d	
27	JMC1364	0.03	0.10	0.27	42.9 ± 0.61	d	

Variation in resistance and physiological traits

In the PCA for the 12 resistance and gas-exchange traits evaluated, five main components were significant with values >1. These components together explained 86.72% of the variation. PC1 explained 32.45% of the total variation in the original data, PC2 18.32%, PC3 15.20%, PC4 11.02% and PC5 explained 9.72% (Fig. 4, Table 3). PC1 consisted of leaf area, SLA, number of leaves and dry mass; PC2 consisted only of PN, PC3 of WUE, PC4 of gs and Ci, and PC5 was formed solely by trichome density (Fig. 4, Table 3).

Figure 4 Biplot of principal component analysis of 27 Phaseolus lunatus landrace accessions from southeastern Mexico based on 12 morphological and physiological leaf traits.

(A) PC1 and PC2, (B) PC2 and PC3. Morphological traits: trichomes (Tr), thickness (Th), hardness (Ha), dry mass of leaves (Dm). Physiological traits: photosynthesis assimilation rate (PN), water-use efficiency (WUE), stomatal conductance (gs), transpiration (E), intercellular carbon (Ci).

Table 3 Variance explained in PCA by five principal components derived from 12 leaf traits of lima bean (P. lunatus) and contributions of the original variables to each component.

Specific leaf area (SLA), photosynthesis assimilation rate (PN), stomatal conductance (gs), intercellular carbon (Ci), water-use efficiency (WUE), transpiration (E), days after emergence (DAE).

	Principal component axes	
Axes	PC1	PC2	PC3	PC4	PC5	
Eigenvalue	3.895	2.199	1.825	1.323	1.167	
Explained proportion of variation (%)	32.456	18.322	15.206	11.026	9.722	
Cumulative proportion of variation (%)	32.456	50.778	65.984	77.010	86.732	
Trait	Correlation matrix	
Leaf area (cm2)	0.974	−0.053	0.093	0.033	−0.056	
Dry mass (g)	−0.776	−0.011	0.267	−0.221	−0.042	
SLA (cm2 g−1)	0.867	−0.085	0.161	−0.06	−0.036	
Trichomes (cm2)	−0.072	−0.023	0.023	−0.078	0.961	
Hardness (g cm−2)	0.454	0.656	−0.129	−0.18	0.182	
Thickness (mm)	−0.389	−0.162	0.62	−0.123	−0.486	
Number of leaves	0.914	0.092	0.162	0.025	0.086	
PN [µmol (CO2) m−2 s−1]	−0.283	0.911	−0.14	−0.042	−0.097	
gs [mol (H2O) m−2 s−1]	0.207	0.184	0.248	0.846	0.146	
Ci [µmol (CO2) mol−1]	−0.091	−0.291	−0.044	0.87	−0.24	
WUE [µmol (CO2) mmol (H2O)−1]	−0.333	0.248	−0.838	−0.157	−0.107	
E [mmol (CO2) mol (H2O)−1]	0.058	0.612	0.667	0.349	0.017	

Association of resistance traits and physiology to performance

The RDA showed a reduced separation of the morphological and physiological variables (eigenvalues axis 1 < 0.1; cumulative variance 99.1%), although the axes were marginally significant (axis 1: F = 2.1, p = 0.054; all axes: F = 1.06, p = 0.056) (Fig. 5). In addition, the damage index was significantly higher (Monte Carlo test, F = 2.04, p = 0.02) in accessions JMC1339 and JMC1288, which had the most leaves and greatest SLA and foliar area. Although the difference in seed yield was not significant, it did tend to be higher in these accessions (Fig. 5). In contrast, JMC1270, JMC1245 and JMC1254 had the least damage, but the lowest seed yield. In addition, they had the highest values of foliar biomass, for gas-exchange variables (PN, E, gs, WUE and Ci) and defense traits (hardness, thickness, and trichome density).

Figure 5 Redundancy analysis showing the ordination of morphological and physiological leaf traits associated with the cumulativedamage index and seed yield of 27 landraces accessions of Phaseolus lunatus from southeastern Mexico.

Accessions: 1 = RRS0001, 2 = JMC1271, 3 = JMC1280,4 = RRS0002, 5 = JMC1255, 6 = JMC1304, 7 = JMC1240, 8 = JMC1350, 9 = JMC1254, 10 = JMC1327, 11 = JMC1273, 12 = JMC1357, 13 = JMC1345, 14 = JMC1270, 15 = JMC1337, 16 = JMC1245, 17 = JMC1208, 18 = JMC1348, 19 = JMC1339, 20 = JMC1288, 21 = JMC1306, 22 = JMC1264, 23 = JMC1325, 24 = JMC1297, 25 = JMC1313, 26 = JMC1336, 27 = JMC1364. Morphological traits: trichomes (Tr), thickness (Th), hardness (Ha), dry mass of leaves (Dm). Physiological traits: photosynthesis assimilation rate (PN), water-use efficiency (WUE), stomatal conductance (gs), transpiration (E), intercellular carbon (Ci).

Discussion

The data showed a wide variation in the evaluated foliar characteristics, in line with the high diversity among the evaluated landrace accessions found by Ballesteros (1999) and Martínez-Castillo et al. (2004) for P. lunatus in the Yucatan Peninsula. In the search for pest resistance among select landrace accessions, genetic variation is a key element because the wider genetic pool increases the likelihood of finding highly resistant populations as found for Vigna umbellata resistant to Callosobruchus chinensis (L.) (Somta et al., 2008), P. vulgaris resistant to Callosobruchus chinensis (Ku-Hwan et al., 2002), and Pisum fulvum resistant to Bruchus pisorum (Clement, Hardie & Elberson, 2002). The effectiveness of species belonging to Fabaceae in resisting damage caused by pest insects is likely a function of multiple defense mechanisms ranging from morphological characteristics to physiological adaptations (Bonte et al., 2010).

When evaluating the resistance traits independently, we found distinct differences among the accessions as reported for other traits of native and cultured materials, thus allowing selection of populations with desirable defense characteristics (Maag et al., 2015; Moya-Raygoza, 2016; Dos Santos et al., 2020). Although many studies have evaluated physical resistance traits, chemical defenses, and biological interaction networks in agricultural production systems, these factors can also contribute to germplasm selection and crop improvement (Chen et al., 2008; Mitchell et al., 2016). Some accessions with larger leaves are likely to have greater vigor and tolerance to damage by herbivorous insects (Ssekandi et al., 2016). Likewise, the thickness and hardness of leaves also have an important impact on resistance; as leaf thickness increases, some sucking insect larvae spend less time on the leaf blade (Rao, 2002; Saheb et al., 2018). To produce large, thick leaves, the plant must also have a greater photosynthetic capacity to generate the necessary photoassimilates. Furthermore, stomatal conductance and transpiration rates play a fundamental role in the WUE of plants with large leaves such as those of P. lunatus. Here, the accessions with outstanding physiological traits such photosynthetic assimilation also had outstanding morphological characteristics for resisting insect damage. Plants with harder leaves require greater effort by leaf-eating insects (Schofeld et al., 2011). Accession JMC1325 in our study conditions produced the hardest and thinnest leaves, so it is a good candidate for further studies on resistance to foliar damage since some plants with thinner leaves are less preferred by insects, as in the case of Vigna radiata (L.) (Lakshminarayan, Singh & Mishra, 2008), V. mungo (Taggar & Gill, 2012), Gossypium hirsutum (Butter & Vir, 1989) and Cucumis sativus (Shibuya et al., 2009). Leaf hardness has been positively associated with the nutritional quality of the leaf; thus, the insects may be able to evaluate and select their food (Larcher, 2006). Therefore, the harder leaves of some accessions might not always be defensive traits, but rather provide a better food source for certain insects, making the plant more susceptible to defoliation (Schädler et al., 2007; Caldwell, Read & Sanson, 2016). In addition, we found that for another widely studied morphological trait, trichome density, accession JMC1306 has a high density. This trait and its contribution to plant defense is difficult to generalize among plant species (Dos Santos et al., 2020). For example, in Fabaceae species, trichomes can contribute to resistance against defoliating insects (Oghiakhe, Jackai & Makanjuola, 1992; Veeranna & Hussain, 1997) by hindering pest movement on the plants (Tian et al., 2012; Figueiredo et al., 2013).

Physiological traits also varied among accessions. Groups of accessions with high values for a particular trait were found, for example, for PN with 13 accessions; for E, Ci and WUE with two; and for gs with only one variable. For PN and E, the evaluated accessions had considerably high levels (PN = 24 PN [µmol (CO2) m−2 s−1] and E = 11 [mmol (H2O) m−2 s−1] (Ribeiro et al. (2004), since values ranged from 25 to 29 [µmol (CO2) m−2 s−1] for PN and 8 to 9 [mmol (H2O) m−2 s−1] for E. Thus, accessions with higher PN seem to have a greater carboxylation capacity in the environmental conditions of the region, and in the case of accessions with high E values, a greater release of water molecules as result of large stomatal openings (Meneses-Lazo et al., 2018). In addition, in the case of gs, most accessions have low values, which could be interpreted as an indicator of drought tolerance (Khazaei et al., 2019), although sometimes the ability to regulate gs can be a better strategy than having low values, as found for Phaseolus vulgaris L. (Rosales et al., 2012). For Ci, we only found two accessions at the highest levels, which could be the result of a differential photosynthetic adjustment between accessions, and in the ability to regulate mitochondrial respiration such as its photorespiration, impacting the release of CO2 (Lawlor & Cornic, 2002). For WUE, accessions JMC1337, JMC1245, and JMC1270 had higher values than the rest and large leaves, a desirable combination of traits for insect resistance.

For all traits described, both resistance and physiological, we highlighted those with the most appropriate values for optimal performance, depending on the type of trait, but an accession should not be selected on the basis of one variable (Capblancq et al., 2018; Vangestel et al., 2018). In this regard, seven accessions—JMC1339, JMC1288, JMC1264, JMC1325, JMC1208 and JMC1313—had the lowest damage indices at 60 DAE. On the basis of leaf damage, they could have a better resistance throughout their ontogeny (Nzungize et al., 2012), but when we also consider higher seed yield, we found that five accessions—JMC1325, JMC1288, JMC1339, JMC1208 and JMC1264—were the best performers. These accessions had the highest yields (67.0 ± 0.92 to 51.9 ± 0.92 g plant−1) and lowest damage indices (0.02 to 0.03). However, when we analyzed with the RDA all accessions simultaneously with the morphological and physiological traits and cumulative damage index during the experiment, we found that only JMC1339 and JMC1288 maintained an optimum seed yield (54.9 ± 0.92 to 55.9 ± 2.24 g plant−1), despite having a high cumulative damage index. Interestingly, we found a strong positive association of leaf production, SLA and leaf area with the cumulative damage index. Overall, this finding may not appear to be very surprising since some plants can maintain high yields while being more susceptible to damage by pests (Lale & Kolo, 1998; Kimiti, Odee & Vanlauwe, 2009; Keneni, Bekele & Imtiaz, 2011; Kiptoo et al., 2016), perhaps because plants that allocate more resources to defense will have less to allocate toward growth or reproduction (Gong & Zhang, 2014). However, we emphasize the association of a higher cumulative damage index with greater leaf production, SLA and leaf area because SLA has been suggested as critical driver of variation in resource availability above ground (Poorter et al., 2012), which might help lima bean compensate for the resource limitation caused by the lost leaf area or even maximize light capture area (high SLA) (Evans & Poorter, 2001; Freschet, Swart & Cornelissen, 2015) through unfolding its leaves in such a way as to avoid leaf overlap (Santiago & Wright, 2007).

Our main results revealed positive correlations between foliar resistance traits and herbivorous insect damage levels but a negative correlation between these traits and yield. Nevertheless, we measured only 45 individuals for each of 27 accessions in only one environment, and the genetic diversity among the accessions has not been assessed. Although it is difficult to generalize the strength of the resistance traits and their possible correlations with insect damage and bean yield, our study highlights the importance of morphological traits such as greater leaf production, leaf area and SLA in relation to increased plant productivity (McNickle & Evans, 2018) through the capture of more light energy and efficient use of available resources in plants (Maschinski & Whitham, 1989).

Conclusions

Our measurements and comparisons of resistance, physiological and yield traits in landrace accessions of P. lunatus in the Yucatan Peninsula highlights the great diversity in germplasm resources. Five accessions, JMC1325, JMC1288, JMC1339, JMC1208 and JMC1264, performed the best in the field in terms of seed yield and lowest cumulative damage index, even though two, JMC1339 and JMC1288, had the greatest damage. Our results found a positive correlation between high values for “resistance traits” and actual resistance to herbivorous insect damages, but a negative correlation between these traits and yield. Our study identifies important morphological (number of leaves, leaf area, and dry mass of leaves; trichome density, specific leaf thickness and hardness) and physiological traits (photosynthetic rate, stomatal conductance, intercellular carbon, water-use efficiency and transpiration) for selecting lima bean accessions belonging to landrace accessions with high yields (regardless of the damage they may suffer) when no agrochemicals are used, despite the limitations of our study. This is the first step toward identifying resistant lines of lima beans for sustainable, safe production in the Yucatan Peninsula.

Supplemental Information

Supplemental Information 1 Raw measurements

The raw data are arranged by pages and each corresponds to an analysis of the article.

Click here for additional data file.

The authors thank Ángel Manuel Herrera Gorocica, Leticia Osalde Navarrete and Ramon Vela Solís for their support in fieldwork.

Additional Information and Declarations

Competing Interests

Author Contributions

Data Availability

The authors declare there are no competing interests.

Roberto Rafael Ruiz-Santiago conceived and designed the experiments, performed the experiments, analyzed the data, prepared figures and/or tables, authored or reviewed drafts of the paper, and approved the final draft.

Horacio Salomón Ballina-Gómez conceived and designed the experiments, analyzed the data, prepared figures and/or tables, authored or reviewed drafts of the paper, and approved the final draft.

Esau Ruiz-Sánchez conceived and designed the experiments, authored or reviewed drafts of the paper, and approved the final draft.

Jaime Martínez-Castillo, René Garruña-Hernández and Rubén Humberto Andueza-Noh performed the experiments, authored or reviewed drafts of the paper, and approved the final draft.

The following information was supplied regarding data availability:

The raw measurements are available in the Supplementary File.

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
