# Peer review of "Determining relevant traits for selecting landrace accessions of Phaseolus lunatus L. for insect resistance"

_PeerJ, doi:10.7717/peerj.12088_

## Round 0.1 · original submission · Major Revisions

· Academic Editor

Major Revisions

A major revision is requested.

·

Basic reporting

English language is professional and clear, even though some sentences should be checked:
L29: “even though” is used while the two accessions cited are not in the genotypes listed in the same sentence (L27 to L28).
L80 and L97 : “data were” should be replaced by “data was”


Introduction is very short and barely explains the background and the question this paper tries to answer. The information of the link between the measured traits and the resistance to insect should be stated in the introduction to explain better how these measures can answer the question. Nevertheless, to my knowledge, this is the first time the correlation between “resistance traits” listed in this paper and damage severity from herbivorous insect is assessed on Phaseolus lunatus. This statement should be placed in the introduction, to clarify the goal of the study.
Nevertheless, we can find information about the link between resistance traits and damage severity in the discussion. I suggest these paragraphs are moved in the introduction:
- Sentence from L197 to L199.
- Sentence from L207 to L212
- Paragraph from L216 to L219


In general, the literature is well referenced and relevant, even though the manuscript will be clearer if the authors mention the model used in the cited papers, to better understand if the knowledge refers to beans, Fabacea, or other crops. In general also, the literature cited is old and the authors should check if newer relevant literature exists.
Here are some specific points:
L38, the Panda and Kush 2001 citation is Panda and Kush 1995 in the reference list, and Chen et al 2015 is Chen et al 2016 in the reference list.
L46, a citation should be added to the statement of beans being the second most important crop in the world
L 199, Saheb et al 2008 is Saheb et al 2018 in the reference list.
L218 Oghiakhe et al 1993 is 1992 in the reference list.


Structure of the article is conform to PeerJ standards and discipline norm.



Figure 1 and 2 would be read more easily if they had a letter to refer to the specific measure. I suggest the graph for “Number of leaves” measure is called “Figure 1 A” and “leaf area” is called “Figure 1 B”, etc. It would also facilitate the citation of these figures in the main text.
- Figure 1 is relevant, even though I suggest the quality of the image should be improved. The bars should be clearly defined (standard deviation, standard error to the mean?)
- Figure 2 is relevant, but the statistical groups of the photosynthetic rate and intercellular carbon are not accurate. For example, on the figure of the photosynthetic rate, genotype 9 is in group “a” while we can clearly see the mean is lower than genotype 3, which is in group “b”. The same issue is found on the figure of intercellular carbon. The statistical analysis should be double-checked. The bars should be clearly defined (standard deviation, standard error to the mean?)
- Figure 3 is relevant but there is no label on the axis. They should state RDA axis number and the percentage of variation which is explained by each axis.
- Table 2 would be read more easily if it was a figure. I suggest a graph format such as a histogram.
- Table 3 would also be read more easily if it was a figure. I suggest data plots with PC1 versus PC2 as axis, and PC2 versus PC3 in another data plot, etc. The title of this figure reports 13 leaf characteristics while there are only 12.
- In supplementary data 1, sheet “Traits” the plot names are still in Spanish

Experimental design

The authors provided an original primary research within scope of the journal.

The question is not well defined, even though it is possible to understand it after reading the manuscript. It should be clearly stated at the end of the introduction, with the knowledge gap of finding a correlation between resistance trait and damage severity on lima bean.


Investigation seems to have been performed rigorously, even if it is not stated how the 15 individual plants were selected per accession, within the 100 available individuals. If they have been selected randomly, it should be stated, and if a method of selection was applied, it should be stated too. Additionally, the authors did not state if the measures were done in a blind manner. If it is not the case, I suggest it should be mention.


Methods are described with sufficient detail in general. Nevertheless, in the statistical analyses section, L96 to L99, it would be clearer if the exact name of the analysis was provided.
The year(s) of the experiment should be added in the “site conditions and crop establishment” section.
L80 to L81, the authors state data was recorded every 15 days from 15 DAE for 3 months, while L71 they state it was at 60 DAE, and at 30, 45 and 60 DAE for insect damage L88. It is hard to understand what measures have been done every 15 days.
L72, it is not clear if the 5 plants evaluated per subplot are the same 5 plants for each trait.
L89, it is not clear if the “same 15 plants selected per plot” are the same plants between the 3 different time points, or the same 15 plants as L72 for the resistance trait measures.

Validity of the findings

Almost all underlying data have been provided. I suggest the means and standard deviation/standard error to the mean values should be provided too. Raw data of Figure 3 should be provided. The statistical analyses of Figure 2 should be double-checked.
The conclusion is not well stated. To my understanding, the authors found a positive correlation between high value in “resistance traits” and actual resistance to herbivorous insect damages, but a negative correlation between these traits and yield. This statement should appear in the conclusion or at the end of the discussion, as it is a great finding for Lima bean breeding. I suggest the author discuss more about the correlation between the traits and the redundancy analysis to understand better the answer to the question.
Because in the material and methods it is unclear if always the same 15 individual plants per accession were used to measure the different traits, it is unclear to conclude about accurate correlations between the different traits. Indeed, the variation of the values is high for some traits, especially number of leaves, leaf area, dry mass of leaves, thickness, SLA, photosynthetic rate, transpiration and WUE. It means that each individuals can differ from another, which shows that measuring different individuals is not an optimal experimental design to conclude about correlation between these traits. Nevertheless, the measures were done only on 15 individuals, in only one environment, and only on 27 accessions for which the genetic diversity has not been assessed. For yield, the number of phenotyped accessions is only 5, leading to a very uncertain correlation since the results might not be representative and repeatable. I suggest a paragraph stating about these limits, in the discussion, should be added.

Additional comments

The authors have measured several morphological and physiological traits, and yield in field on Lima bean in Mexico. They evaluate the different correlations that can be made thanks to these measures.
In general, the manuscript is very descriptive and it is hard to understand what conclusions the authors want to draw. Even though the descriptions are relevant, it is hard to understand what is the question, and how it is answered. As a summary, here are the major reviews:
- The redundancy analysis should be discussed more to understand better the answer to the question
- The question should be stated at the end of the introduction
- The statistical analysis of Figure 2 should be checked
- The limits of the approach in terms of representativity of the accessions and repeatability of the results should be stated
- The conclusion should answer the question stated at the end of the introduction
- The results of table 2 and table 3 should be shown in a figure format
Nevertheless, here are some additional minor revisions:
In the results section, the citations of the highest and lowest values of each measure makes the reading of the manuscript hard. In general, there is no consistency in the order in which they are cited, which is sometimes by sorting them with the value of the measure, sometimes in the sorted order from Figure 1 and Figure 2, and sometimes with no apparent structure. Additionally, the citations of the relevant genotypes in the Abstract and the Results section are often mixed up. They should be double-checked, but here are the ones I have found:
- L29, JMC128839 is cited twice and does not exist, they should be replaced by JMC1339 and JMC1288
- L122, JMC1271 is cited as being the highest value leaf dry mass, while it is not the case.
- L126, JMC1325 is cited as carrying the thickest leaves, while it should be JMC1255
- L133 to L135, the genotypes cited as having the highest and lowest photosynthetic rates don’t match the figure 2 and the raw data.
- L137 the JMC1364 is not supposed to be cited as the highest photosynthetic rate, it should be JMC1339
- L137, JMC1327 should be cited instead of JMC1364
- L139 and L140, JMC1245, JMC1337 should be replaced by JMC1350 and JMC1288. JMC1336 should be replaced by JMC1339

Reviewer 2 ·

Basic reporting

The paper describes interesting research on the possibility of introducing native forms of Phaseolus lunatus L. beans into cultivation and their sensitivity to damage by herbivorous insects.
The aim of this study was to select accessions that are most resistant to damage caused by defoliating insects from among 27 local accessions of lima beans (P. lunatus) by characterizing morphological, physiological and yield traits known to contribute to insect resistance such as dry mass of leaves, leaf area, specific leaf area (SLA), leaf thickness and hardness, and trichome density, improvement in photosynthetic capacity, and its impact on yield.
The layout of the work is correct. Divided into introduction, research methods, results, discussion and conclusions. At the end of the work, there is a references. The test results are presented in 3 tables and 3 figures. The abstract is correctly written. It contains a summary of the conducted research and results. The statistical methods were performed correctly. The conclusions correspond to the purpose of the work. The cited literature is correct.

Experimental design

However, errors were found in the work that need to be corrected
1. There is no research hypothesis in the introduction, it should be added.
2. In materials and methods (site conditions and crop establishment), the soil conditions and the course of the weather during the research period and the applied plant fertilization should be added.
3. In the data collection chapter (line 77, add the method by which the SLA was marked)
4. Research results need to be improved. They are difficult to read due to the large number of digits. I propose to include statistical indicators (f, Df, p) in the table, which will improve the transparency of the work. The authors define a higher -lower level for all examined features, which results from tables and charts (maybe it should be specified by how many% a given feature is higher or lower). In the chapter (Physiological traits) line 137 is JMC 1364 and it should be 1339. Other data of physiological traits are also inconsistent with the data in figure 2. They should be corrected
5. The discussion of the results is not fully understandable and requires linguistic improvement
6. The literature does not cite Mitchell and el. (line 374)
7. In table 2 it is doubtful that the seed yield of JMC 1348 (62.0 g / plant-1) and JMC1325 (62.5 g / plant-1) belongs to 2 different homogeneous groups (a and b).
8. In table 3 (line 3) it should be 12 and not 13.
9. Correction is required in English by a professional translator.

Validity of the findings

The work is an original scientific research, it brings new elements to agricultural research. The proper statistical analysis of the research results deserves attention.
After detailed corrections (minor revision), the work can be published in PeerJ.

Reviewer 3 ·

Basic reporting

The introduction needs more detail. I suggest that Authors should improve the description at lines 46-50 to provide more justification for the study e.g. Why were beans selected for testing?
What pests attack him?
What are the problems in growing, etc.

Experimental design

The methods are insufficiently described.
There is no description of the conditions for growing beans.
What was the soil type?
What was the soil abundance in nutrients?

Was WUE measured with the gas analyzer?

Authors should describe the scale by which leaf damage was assessed.

Validity of the findings

The description of the results does not agree with the data presented in the table 2 and figures 1 and 2. There are numerous errors in the description of the research results and the homologous groups in the fig. 2 (e.g. photosynthesis).

Besides, when describing the results, there should be numbers next to the bean accessions, the same as in the figures (in parentheses), because data analysis is very difficult.

More comments are in manuscript.

Additional comments

The manuscript is interesting and contains a lot of valuable information, but I found a lot of mistakes that disqualify this paper in current form.
The work requires checking and correcting the entire results section and Introduction as well as Material and Methods sections supplements.

Annotated reviews are not available for download in order to protect the identity of reviewers who chose to remain anonymous.

---

## Round 0.2 · Minor Revisions

· Academic Editor

Minor Revisions

Dear Authors,

The MS still needs some revision.

The units of plant gas exchange parameters should be corrected in the whole text and figures legends:

Change all An to PN

then consider these units:

PN- (µmol CO2 m-2 s-1)
E (µmol H2O m-2 s-1)
gs - (µmol H2O m-2 s-1)
Ci (µmol CO2 m-2 s-1)
WUE (µmol Co2 mmol H2O-1)

See this link for SE units and abbreviations:

https://ps.ueb.cas.cz/incpdfs/inf-990000-0200_10_116.pdf

---

## Round 0.3 · Minor Revisions

· Academic Editor

Minor Revisions

Please address the following comments from the Section Editor:

> This appeared a straightforward analysis of germplasm diversity under insect pressure. The objectives were to evaluate the germplasm apart from the general yield parameters under pesticide control measures. This may be a first step in evaluating germplasm parameters important for insect resistance; however, to be of value to the reader it would be important to characterize the diversity in a fashion more useful than that in Table 1 using molecular tags to develop its station among other known germplasm.
>
> As this is a select set, how would the reader be able to conduct similar tests and be able to compare their results here; was there a standard germplasm here which is readily available?
>
> Or how would a reader be able to obtain the germplasm needed to benchmark a study against this? In general the manuscript was written well and is acceptable, but some way to tie the study into a context from which the reader would be able to validate the study with a reference selection would have been preferred. Perhaps one of the accessions has a pedigree which fits the bill and could be mentioned.

---

## Round 0.4 · accepted · Accept

· Academic Editor

Accept

In the Materials & Methods section, when describing WUE please change the abbreviation of assimilation rate (AN) to photosynthetic assimilation rate (PN).